# Goethite and Hematite Nanoparticles Show Promising Anti-Toxoplasma Properties

**DOI:** 10.3390/pharmaceutics16030413

**Published:** 2024-03-18

**Authors:** Kosei Ishii, Eiji Akahoshi, Oluyomi Stephen Adeyemi, Hironori Bando, Yasuhiro Fukuda, Tomoyuki Ogawa, Kentaro Kato

**Affiliations:** 1Laboratory of Sustainable Animal Environment, Graduate School of Agricultural Science, Tohoku University, 232-3 Yomogida, Naruko-onsen, Osaki 989-6711, Miyagi, Japan; kousei.ishii.p1@dc.tohoku.ac.jp (K.I.); oluyomiadeyemi@gmail.com (O.S.A.); hironori_bando@asahikawa-med.ac.jp (H.B.); yasuhiro.fukuda.b7@tohoku.ac.jp (Y.F.); 2Department of Electronic Engineering, Graduate School of Engineering, Tohoku University, 6-6-05 Aza-Aoba, Aramaki, Aoba-ku, Sendai 980-8579, Miyagi, Japantomoyuki.ogawa.d1@tohoku.ac.jp (T.O.); 3Department of Biochemistry, Medicinal Biochemistry, Nanomedicine & Toxicology Laboratory, Bowen University, Iwo 232101, Osun State, Nigeria; 4Department of Parasitology, Asahikawa Medical University, 2-1-1-1 Midorigaoka-Higashi, Asahikawa 078-8510, Hokkaido, Japan

**Keywords:** nanoparticle, parasite, tryptophan

## Abstract

*Toxoplasma gondii* is an intracellular parasitic protozoan with a high infection rate in mammals, including humans, and birds. There is no effective vaccine, and treatment relies on antiparasitic drugs. However, existing antiprotozoal drugs have strong side effects and other problems; therefore, new treatment approaches are needed. Metal nanoparticles have attracted increased interest in the biomedical community in recent years because of their extremely high surface area to volume ratio and their unique reactivity that could be exploited for medicinal purposes. Previously, we confirmed the anti-Toxoplasma effects of gold, silver, and platinum nanoparticles, in a growth inhibition test. Here, we asked whether the anti-Toxoplasma effect could be confirmed with less expensive metal nanoparticles, specifically iron oxide nanoparticles (goethite and hematite). To improve the selective action of the nanoparticles, we modified the surface with l-tryptophan as our previous findings showed that the bio-modification of nanoparticles enhances their selectivity against *T. gondii*. Fourier-Transform Infrared Spectroscopy (FTIR) analysis confirmed the successful coating of the iron oxide nanoparticles with l-tryptophan. Subsequently, cytotoxicity and growth inhibition assays were performed. L-tryptophan-modified nanoparticles showed superior anti-Toxoplasma action compared to their naked nanoparticle counterparts. L-tryptophan enhanced the selective toxicity of the iron oxide nanoparticles toward *T. gondii*. The bio-modified nanoparticles did not exhibit detectable host cell toxicity in the effective anti-Toxoplasma doses. To elucidate whether reactive oxygen species contribute to the anti-Toxoplasma action of the bio-modified nanoparticles, we added Trolox antioxidant to the assay medium and found that Trolox appreciably reduced the nanoparticle-induced growth inhibition.

## 1. Introduction

*T. gondii* is a protozoan parasite that infects most birds and mammals and causes the zoonotic disease toxoplasmosis. In the life cycle of *Toxoplasma*, oocysts expelled from the terminal host cat by feces or other means enter the bodies of mammals and birds, where they transform into tachyzoites [1]. *T. gondii* infects more than one-third of all humans and is asymptomatic in healthy individuals but causes severe symptoms in the immunocompromised [2,3,4]. In addition, if a pregnant woman is infected with *T. gondii*, it can cause in utero infection, resulting in miscarriage or birth defects [5,6,7]. It is also the second leading cause of death from food poisoning [8], resulting in many economic and human losses worldwide. There is no effective vaccine against *T. gondii*, and the treatment of infections caused by this protozoan is dependent on antiparasitic drugs. Pyrimethamine, one of the drugs used to treat toxoplasmosis, is also known to have side effects [9]. Therefore, a new approach to treatment is required. 

We have focused on metal nanoparticles (NPs), which have attracted increased interest in the biomedical community [10], and have analyzed their antiparasitic effects against *T. gondii*. Metal nanoparticles have been the subject of various biological studies due to their characteristics. Examples include drug carriers for drug transport due to their easy surface modification and targeting specific molecules in cells due to their high light scattering and absorption efficiency. Experiments using metal nanoparticles as a therapeutic agent for parasitic infections have also been conducted for other parasites as well, indicating the potential of metal nanoparticles [11]. Metal NPs exhibit different properties than their bulk counterparts due to their large surface area relative to their volume [12]; therefore, they are thought to stimulate the generation of reactive oxygen species in cells and exhibit parasitic activity [13,14,15]. The small size of NPs also allows them to penetrate cell membranes, making them highly reactive. Metal NPs are already being used for biomedical applications [16,17,18]. Although it has been shown that gold, silver, and platinum metal NPs exhibit antiparasitic activity [19,20,21,22], these are precious metals and are expensive. In this study, we focused on iron, an inexpensive metal, and asked whether iron oxide NPs exhibit antiparasitic activity against *T. gondii*. Iron oxide nanoparticles have a variety of medical applications, including MRI and cancer therapy [23]. It has also been reported to have no significant side effects under certain conditions [24], leading to this verification as a candidate for an anti-Toxoplasma drug. We examined three relatively stable iron oxides: goethite, hematite, and magnetite. Previous studies have also shown that the L-tryptophan coating of metal NPs increases their anti-Toxoplasma effect [25,26]. Since *Toxoplasma* has a tryptophan requirement [27,28,29,30], we also investigated whether the l-tryptophan modification of iron oxide NPs enhances the anti-Toxoplasmic effect.

## 2. Materials and Methods

### 2.1. Materials

Goethite NPs (FeO(OH)NP, 60 nm), hematite NPs (Fe_2_O_3_NP, 50 nm), and magnetite NPs (Fe_3_O_4_NP, 50–100 nm) were purchased from Future Materialz (Tokyo, Japan). L-tryptophan and bovine serum albumin (BSA) were purchased from Sigma-Aldrich (St. Louis, MO, USA). Trolox (6-hydroxy-2,5,7,8-tetramethylchroman-2-carboxylic acid) was purchased from Santa Cruz Biotechnology Inc. (Dallas, TA, USA).

### 2.2. Methods

#### 2.2.1. Parasites

We used *T. gondii* RH strain 2F in this study. The parasite was maintained by repeated passages in monolayers of Vero cells (American Type Culture Collection, Manassas, VA, USA) cultured in Dulbecco’s Modified Eagle’s medium (DMEM; Nacalai Tesque, Kyoto, Japan) supplemented with 5% (*v*/*v*) fetal bovine serum (FBS) and penicillin and streptomycin (100 U/mL; Thermo Fisher Scientific Inc, Waltham, MA, USA). Host cells infected with *T. gondii* tachyzoites were passed through a 27 G needle to lyse them. The cell lysates were then filtered through a 5 µm filter to obtain a tachyzoite suspension free of host cell debris. The suspension was centrifuged (400× *g*, 10 min, 23 °C), and the supernatant was removed and suspended in fresh culture medium. Then, the parasite density was measured by using a hemocytometer and adjusted for in vitro experimental infection analyses.

#### 2.2.2. Cytotoxicity of Metal NPs in Mammalian Cells

By using previously reported methods [16], we maintained HFF cells in DMEM supplemented with 5% (*v*/*v*) FCS and penicillin and streptomycin (100 U/mL). Cells were grown to confluence at 37 °C in a 5% CO_2_ atmosphere. All experiments were performed in 96-well plates (Nunc) unless otherwise stated. At confluence, cells were trypsinized and resuspended to the desired cell density. The cells were seeded onto plates at a density of 1.0 × 10^4^ cells/well and incubated for 48 h followed by treatment with various concentrations (between 0.01 and 100 µg/mL) of the NPs (goethite NPs, hematite NPs, and magnetite NPs). Culture medium lacking the test compounds was added to the control well, and the medium-only well was used to correct for any background signal. The treated cells were incubated for 48 h before being subjected to the cell viability assay. 

Cell viability was determined using the CellTiter 96^®^ AQueous One Solution Cell Proliferation Assay kit (Promega, Madison, WI, USA). Briefly, the well plate and its contents were equilibrated to room temperature. Then, 100 µL of the CellTiter 96^®^ AQueous One reagent was added to each well. The contents were briefly mixed on an orbital shaker and then incubated at 37 °C in a 5% CO_2_ atmosphere for 1–4 h. The absorbance signal was recorded at 490 nm using a microplate reader (GloMax Navigator 96; Promega). The assay was repeated three times in triplicate. The results are presented as the mean ± standard error of the mean (SEM; *n* = 3) of three independent experiments.

#### 2.2.3. In Vitro Growth Inhibition Assessment by Use of Luciferase Reporter Assays

The number of *T. gondii* tachyzoites was determined by using a luminescence-based assay of β-galactosidase (β-gal) activity expressed by the parasite strain RH-2F, as described previously [16]. To obtain a purified parasite suspension for the assays, infected cells were syringe-released, and the lysates were filtered to remove cell debris.

The growth inhibition assays and in vitro invasion assays were performed as described elsewhere [26]. For the growth inhibition assay, purified parasite (1 × 10^4^) suspension was added to growing monolayers and invasion was allowed to occur for 1 h. Then, fresh medium containing the NPs (reconstituted in culture medium) was added. The monolayers were then incubated for 48 h. The mock-treated (treated with NP vehicle only; in this case, culture medium) cells served as a positive control, whereas the medium-only well was used to correct for any background signal. After the 48 h incubation at 37 °C in a 5% CO_2_ atmosphere, the viability of the RH-2F parasite strain was determined by measuring the galactosidase expression in a Beta-Glo^®^ Luminescent Assay kit (Promega, Madison, WI, USA). The assay was performed in triplicate and repeated three times. All experiments were performed in 96-well optical bottom plates (Nunc; Fisher Scientific, Pittsburgh, PA, USA) unless otherwise stated.

#### 2.2.4. Tryptophan-Coated Iron Oxide Nanoparticles

Tryptophan was purchased from Sigma-Aldrich (St. Louis, MO, USA). Using previously reported methods [26], powdered tryptophan was added to DMSO and stirred for 10 min, then goethite and hematite were added, and the mixture was stirred for another 10 min. Then, with the support of the ARIM Support Office of Chitose University of Science and Technology, the coating was assessed by the use of FTIR spectroscopy to confirm that it was successful. We determined the concentration of the bio-modified TiO_2_ NPs by the use of a gravimetric method.

## 3. Results

### 3.1. Optimal Nanoparticle Concentration Assessment

First, to test the growth inhibition of *Toxoplasma* at NP concentrations that would not significantly affect host cell viability, toxicity tests were conducted on host cells in the absence of *Toxoplasma* using various iron oxide NPs (Figure 1). We found that the survival rate exceeded 80% at concentrations of 10 μg/mL or less for goethite, hematite, and magnetite, respectively. Therefore, we used concentrations of ≤10 μg/mL in subsequent experiments.

### 3.2. Concentration-Dependent Growth Inhibition by Iron Oxide Nanoparticles

Based on the results obtained from the cytotoxicity test, we conducted a growth inhibition test at NP concentrations that caused ≤20% toxicity. The relative numbers of *Toxoplasma* were significantly reduced at concentrations of ≥10 μg/mL for goethite and ≥1.0 μg/mL for hematite (Figure 2). These results suggest that iron oxide NPs of goethite and hematite may have growth inhibitory effects on *Toxoplasma*.

### 3.3. Surface Modification of Iron Oxide Nanoparticles with Tryptophan

Previous studies have shown that tryptophan coating on gold, silver, and platinum nanoparticles increases their growth inhibitory effect on *Toxoplasma* [31]. Therefore, we applied tryptophan coating to goethite and hematite. FTIR spectroscopic analysis showed that tryptophan-coated goethite and hematite peaked in the same wave number range as tryptophan (Figure 3). This suggests that the iron oxide NPs were coated with tryptophan, albeit in small amounts.

### 3.4. Host Cytotoxicity Testing of Tryptophan-Coated Iron Oxide Nanoparticles

Before testing the inhibitory effect of the tryptophan-coated iron oxide NPs on *Toxoplasma* growth, we tested their toxicity in host cells. Strong toxicity (approximately 90% reduction in cell viability) was observed at a concentration of 50 µg/mL for tryptophan-coated goethite, whereas concentrations up to 2.0 µg/mL of tryptophan-coated hematite resulted in a 20% reduction in cell viability (Figure 4). Based on these results, growth inhibition studies were conducted at concentrations below 25 μg/mL for tryptophan-coated goethite and 2.0 μg/mL for tryptophan-coated hematite.

### 3.5. Tryptophan-Coated Iron Oxide Nanoparticles Increase the Growth Inhibitory Effect of the Nanoparticles on Toxoplasma

To investigate the effect of the tryptophan coating on the growth inhibitory effect of iron oxide nanoparticles, we performed the growth inhibition test using tryptophan-coated nanoparticles at various concentrations less than 50 μg/mL. The relative number of *Toxoplasma* was significantly reduced at a concentration ≥ 12.5 μg/mL of tryptophan-coated goethite NPs. For hematite NPs, the relative number of *Toxoplasma* was significantly reduced at concentrations ≥ 1.0 μg/mL (Figure 5A,B).

In addition, a significant difference in the relative number of *Toxoplasma* was observed for both types of tryptophan-coated iron oxide NPs compared to the same concentration without coating. These results suggest that the tryptophan coating of iron oxide NPs may increase their growth inhibitory effect on *Toxoplasma*.

### 3.6. The Growth Inhibitory Effect of Iron Oxide Nanoparticles May Be Due to the Generation of Reactive Oxygen Species

Prior studies have suggested that the mechanism by which gold, silver, and platinum NPs inhibit growth involves the generation of reactive oxygen species (ROS) [32]. To test whether the same mechanism of action is true for iron oxide NPs, we examined the role of ROS by adding the antioxidant Trolox [33,34] to the *Toxoplasma* growth inhibition assay medium.

Trolox attenuated the growth inhibition induced by tryptophan-coated goethite and hematite (Figure 6A,B). There was no significant difference in the non-tryptophan-coated goethite and hematite to the Trolox group, which may be because the lower concentrations of goethite and hematite did not show growth inhibition effects. These results suggest that ROS may be involved in the growth-inhibiting action of tryptophan-coated iron oxide NPs against *Toxoplasma*.

## 4. Discussion

Metal nanoparticles are known to have antimicrobial effects [19,20,21,22], and this study tested whether they could be applied to *Toxoplasma*. Although previous studies have shown that gold, silver, and platinum NPs exhibit anti-*Toxoplasma* effects [16], these are expensive metals, and therefore their extensive use, including in livestock, is financially probative necessitating the use of NPs of less expensive metals. Here, we confirmed the anti-Toxoplasmic effect and the lack of host cytotoxicity of NPs of iron oxide.

Since damage to host cells by iron oxide NPs (goethite, hematite, and magnetite) could cause strong side effects, we first examined their effect on host cell viability. Cell survival rates in the presence of each iron oxide nanoparticle exceeded 80% at concentrations of >10 μg/mL. Based on these results, we performed a growth inhibition test at concentrations of >10 μg/mL.

To determine the effect of iron oxide nanoparticles on *Toxoplasma* growth, we analyzed the relative survival of *Toxoplasma* after 48 h of incubation with iron oxide NPs in HFFs infected with *Toxoplasma*. We found that the relative number of *Toxoplasma* was significantly reduced at concentrations of ≥10 μg/mL for goethite and ≥1.0 μg/mL for hematite. The EC_50_ could not be calculated because the relative counts were not less than 50% at the concentrations used in this study. These results confirm the growth inhibitory effect of goethite and hematite NPs on *Toxoplasma*. Our results thus confirmed previous studies [16] demonstrating that metal NPs other than those bearing precious metals have anti-Toxoplasma effects. Considering the widespread use of anti-Toxoplasma drugs in humans and livestock, it is significant that an anti-Toxoplasma effect was observed in such a relatively inexpensive metal.

Tryptophan coating on gold, silver, and platinum nanoparticles has been shown to increase their growth inhibitory effect on *Toxoplasma* [31]. Therefore, we thought that tryptophan coating on iron oxide nanoparticles would increase the growth inhibitory effect. We coated iron oxide NPs with tryptophan, and a solution of tryptophan dissolved in DMSO was analyzed by FTIR spectroscopy with the support of the ARIM Support Office of the Chitose University of Science and Technology. The results showed that tryptophan-coated goethite and hematite showed peaks at 2500–3500 cm^−1^, which is the specific wavenumber range of tryptophan. This result means that a small amount of tryptophan coated the surface of the goethite and hematite NPs. The tryptophan-coated iron oxide NPs were used for cytotoxicity testing, and all but 50 μg/mL of tryptophan-coated goethite resulted in survival rates exceeding 80%. To our knowledge, no previous studies have reported increased toxicity with tryptophan coating [26]. Therefore, the reason for the very strong toxicity at 50 μg/mL of goethite in this study is not clear and is a subject for future investigation.

The growth inhibition effect of tryptophan-coated iron oxide NPs was verified, and significant growth inhibition was shown for both goethite and hematite. A comparison of the same concentration of NPs with and without the amino acid coating showed that the l-tryptophan-coated iron oxide NPs exhibited a significant growth inhibitory effect. As reported elsewhere [16], the tryptophan requirement of *Toxoplasma* may contribute to its increased sensitivity to the amino acid-coated NPs. Coating the nanoparticles with l-tryptophan might have led to an increased local concentration in the *Toxoplasma* as the parasite sought to acquire this nutrient from its host. However, it is not clear from the results of this study whether the iron oxide nanoparticles penetrate the parasite, and further analysis is needed to determine the mechanism by which the growth inhibitory effect of *Toxoplasma* was enhanced.

Prior studies have suggested that the mechanism of action of gold, silver, and platinum NPs involves decreasing the mitochondrial membrane potential by generating ROS [32]. We confirmed the effect of adding Trolox, an antioxidant, on the relative viability of *Toxoplasma* in the presence of the NPs. The attenuation of growth inhibition by the addition of Trolox was confirmed in both tryptophan-coated goethite and hematite. There was no significant difference in the non-tryptophan-coated goethite and hematite compared to the Trolox group, which may be because the lower concentrations of goethite and hematite did not show growth inhibition effects. These results suggest that ROS may be involved in the growth-inhibiting action of tryptophan-coated iron oxide NPs against *Toxoplasma*, consistent with previous reports on gold, silver, platinum, and titanium dioxide [2,16]. Previous studies have reported that ROS reduces the membrane potential of *Toxoplasma*, thereby decreasing the production of ATP and causing a growth-inhibitory effect, and it is possible that a similar mechanism of action caused the growth-inhibitory effect [8]. The fact that the addition of Trolox did not result in 100% relative parasite survival suggests that factors other than ROS development may be involved in the growth inhibition effect, which needs to be verified in the future.

## 5. Conclusions

Our data show that iron oxide NPs inhibit the growth of *Toxoplasma* tachyzoites in vitro. Goethite and hematite show promising anti-Toxoplasma properties. Furthermore, the coating of goethite and hematite with l-tryptophan enhances their anti-Toxoplasma action without a corresponding increase in host cell toxicity. Collectively, our findings support the potential of nanoparticles as novel treatment agents for toxoplasmosis. In addition, ROS may be involved in this growth inhibitory effect. The demonstration of the anti-toxoplasmic effect of iron oxide, a relatively inexpensive metal, in this study is very important for the future use of anti-toxoplasmic drugs in a wide range of human and animal species. Future research will aim to further strengthen the growth-inhibitory effect of iron oxide nanoparticles by elucidating the more detailed mechanism of action and clarifying what points affect the growth-inhibitory effect.

## Figures and Tables

**Figure 1 pharmaceutics-16-00413-f001:**
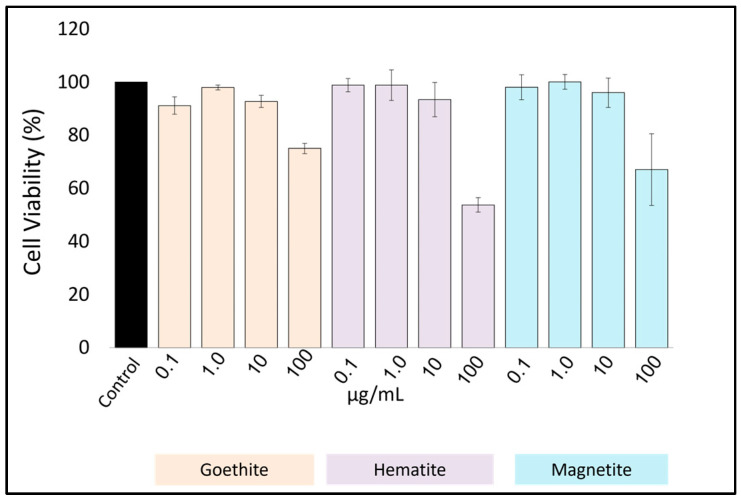
In the absence of *T. gondii* infection, host monolayers were treated with NPs at the effective anti-T. gondii concentration, and cell viability was determined after a 48 h incubation. HFF was seeded at a desired density of 4.0 × 10^4^ cells/well. The experiment was conducted three times independently in triplicate. The data shown are the means ± standard deviation (SD).

**Figure 2 pharmaceutics-16-00413-f002:**
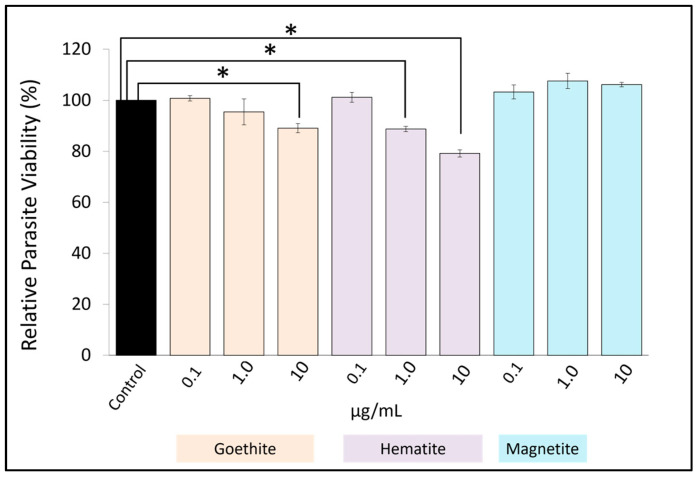
Relative number of *Toxoplasma* parasites after 48 h of infection of HFFs with *Toxoplasma* and addition of iron oxide nanoparticles (goethite, hematite, and magnetite). Figures are triplicates and averages of three independent runs. Data are means ± standard deviation (SD). Experiments were performed in triplicate and repeated three times independently; ns, not significant at *p* > 0.05; *, significant at *p* < 0.05.

**Figure 3 pharmaceutics-16-00413-f003:**
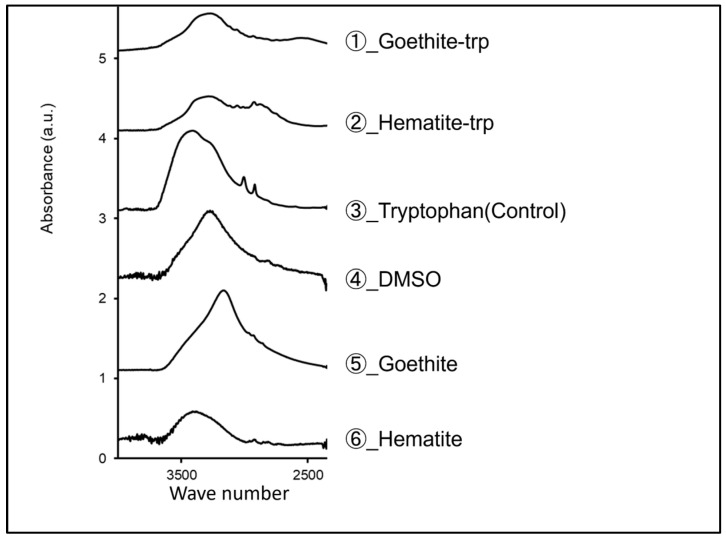
Tryptophan-coated and uncoated iron oxide nanoparticles (goethite and hematite), as well as tryptophan and DMSO, were analyzed by FTIR spectroscopy. The vertical axis shows absorbance, and the horizontal axis shows wavenumber (cm^−1^).

**Figure 4 pharmaceutics-16-00413-f004:**
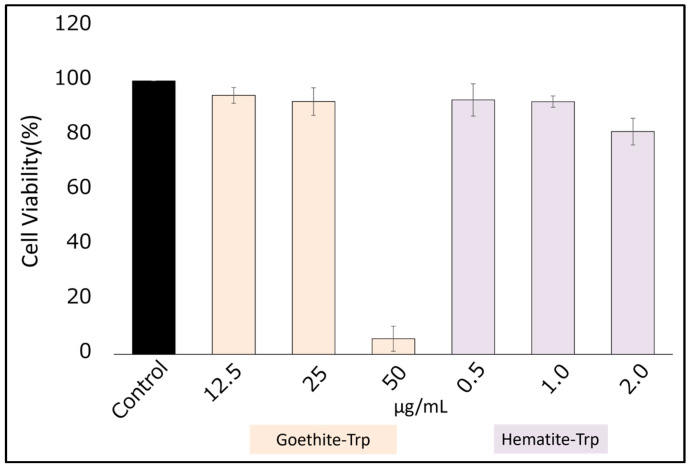
The cytotoxicity testing of tryptophan-coated iron oxide nanoparticles (goethite and hematite). The maximum concentration was adjusted for both goethite and hematite, and three concentrations were prepared by two-fold dilution. Data are means ± standard deviation (SD).

**Figure 5 pharmaceutics-16-00413-f005:**
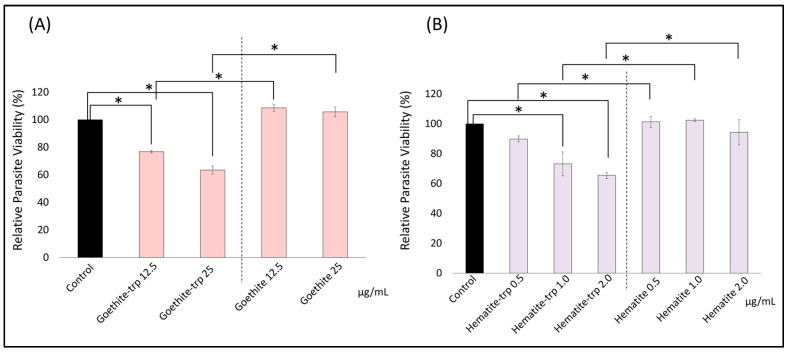
*Toxoplasma* growth inhibition using tryptophan-coated iron oxide nanoparticles. Tryptophan-coated iron oxide nanoparticles were added to Toxoplasma-infected cells, and the relative number of *Toxoplasma* was calculated after 48 h of incubation. (**A**) Results for tryptophan-coated goethite; (**B**) results for tryptophan-coated hematite. Data are means ± standard deviation (SD). Experiments were performed in triplicate and repeated three times independently; *, significant at *p* < 0.05.

**Figure 6 pharmaceutics-16-00413-f006:**
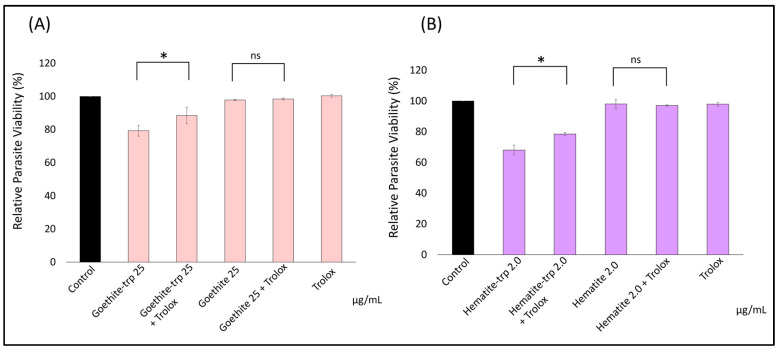
The effect of the antioxidant Trolox on the anti-Toxoplasma effect of iron oxide nanoparticles. Trolox was added after the addition of iron oxide nanoparticles, and the relative survival of *Toxoplasma* after incubation was calculated. The results of (**A**) goethite and (**B**) hematite are shown. Data are means ± standard deviation (SD). Experiments were performed in triplicate and repeated three times independently; ns, not significant at *p* > 0.05; *, significant at *p* < 0.05.

## Data Availability

The data presented in this study are available on request from the corresponding authors.

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
