# Peer review of "Goethite and Hematite Nanoparticles Show Promising Anti-Toxoplasma Properties"

_pharmaceutics, 2024, doi:10.3390/pharmaceutics16030413_

Round 1

Reviewer 1 Report

Comments and Suggestions for Authors

The manuscript “Goethite and hematite nanoparticles show promising anti-Toxoplasma properties” authored by Kosei Ishii and collaborators describes the anti-Toxoplasma activities of iron oxide nanoparticles. Despite the high burden of Toxoplasmosis worldwide, there are no specific treatments for this disease. On the contrary, antiparasitic drugs with strong side effects are currently being used, making it urgent to develop new therapeutic strategies against Toxoplasma. The authors demonstrated that goethite and hematite nanoparticles are able to reduce parasite viability at concentrations below 10 µg/mL. Moreover, L-tryptophan coating of the nanoparticles improved the selective toxicity of the iron oxide nanoparticles towards Toxoplasma gondii. Overall, the manuscript presents an interesting approach that could serve as a potential alternative for combating Toxoplasma. However, certain adjustments and clarifications are required to meet the standards of publication in Pharmaceutics.

Specific comments:

1. Please ensure that scientific names are italicized when writing them. Also, the term "Toxoplasmas" is not correct (line 164), for plural form, the authors can use Toxoplasma parasites.

2. This manuscript could greatly benefit from the inclusion of a brief description of the Toxoplasma life cycle in the introduction section, as well as the number of cases and deaths reported in the last year. Additionally, expanding the current anti-Toxoplasma drug regimens and highlighting the main side effects associated with the treatments is important.

3. In line with my previous comment, it would be important to describe the potential benefits of using metal nanoparticles for Toxoplasmosis treatment, describing the current landscape of these nanoparticles in treating other parasitic diseases.

4. In the results sections it appears to be a miss-connection between the part 3.1 and 3.2. In 3.1 the authors conclude that concentrations below 10 micrograms per mL will be used in further experiments. However, in section 3.2 the authors describe a significant reduction in Toxoplasma growth using goethite NPs at concentrations above 10 micrograms per mL. Please clarify.

5. Include the number of replicates performed in Figure 1.

6. Section 3.4 needs to be rephrased, as the main message of the text is unclear. Upong reviewing Figure 4, it appears that different concentrations of goethite-Trp NPs and hematite-Trp NPs were tested. If that is the case, the conclusion on lines 193-194 is inaccurate, as the highest concentration for Hematite-Trp NPs should be below 2.0 micrograms per mL.

7. The section 3.5 requires further clarification. In my perspective, Figure 5 indicates that non-coated goethite NPs did not reduce the parasite viability at a concentration of 12.5 micrograms per mL. However, in Figure 2, there is a significant reduction in the parasite growth at a lower concentration (10 micrograms per mL). Similarly, uncoated hematite NPs at a concentration of 1.0 micrograms per mL show discrepancies between Figures 2 and 5. This should be addressed by the authors. Moreover, presenting a table showing the means and SD values could significantly enhance the interpretation of the results.

Comments on the Quality of English Language

I kindly request that the authors consider revising the English language in the manuscript to enhance clarity and readability. For instance, line 40 should be towards instead of toward; line 137, was instead of were, etc.

Author Response

Editors-in-Chief

Pharmaceutics

                                                                                                    February 25, 2024

Dear Editors-in-Chief:

              Please find enclosed our revised manuscript (pharmaceutics-2881212), “Goethite and hematite nanoparticles show promising anti-Toxoplasma properties” which we would like to be considered for publication as a primary research article in Pharmaceutics.

The reviewer’s comments and responses are mentioned below for your consideration.

The article is the original work of all authors. The result or part of the result has not been submitted to any journal, has not been published in any journal or not under any consideration to submit in future. All authors have read and approved the final version to submit to your journal, Pharmaceutics. There has no conflict of interest to declare.

Thanks in advance for your cooperation. Eagerly waiting for your response.

Kentaro Kato

Reviewer 2 Report

Comments and Suggestions for Authors

The presented article is devoted to the study of metal nanoparticles (iron oxide nanoparticles) coated with amino acid (tryptophan) as anti-Toxoplasma agents. Such particles have been shown to have a good anti-Toxoplasma effect compared to their analogues from unmodified nanoparticles with their moderate toxicity to host cells at effective anti-Toxoplasma doses. Without a doubt, the article will be of interest to readers of the Pharmaceuticals.

 There are the following comments to the article:

- The introduction of the article is too short. It should be expanded by including references to work on iron oxide nanoparticles, the coating of these particles with various agents and their possible toxicity to the body.

- For iron oxide nanoparticles, only the average size is given. It is necessary to provide more information, for example, how monodisperse these nanoparticles are.

- In future publications, it is necessary to present atomic force microscopy data showing the morphological features of the surface of nanoparticles.

- Line 271: Should be cm-1 instead of m-1. It is also necessary to provide this dimension along the axis representing the wavelengths (Figure 3).

Author Response

Editors-in-Chief

Pharmaceutics

                                                                                                    February 25, 2024

Dear Editors-in-Chief:

              Please find enclosed our revised manuscript (pharmaceutics-2881212), “Goethite and hematite nanoparticles show promising anti-Toxoplasma properties” which we would like to be considered for publication as a primary research article in Pharmaceutics.

The reviewer’s comments and responses are mentioned below for your consideration.

The article is the original work of all authors. The result or part of the result has not been submitted to any journal, has not been published in any journal or not under any consideration to submit in future. All authors have read and approved the final version to submit to your journal, Pharmaceutics. There has no conflict of interest to declare.

Thanks in advance for your cooperation. Eagerly waiting for your response.

Sincerely yours,

Kentaro Kato

Reviewer 3 Report

Comments and Suggestions for Authors

Authors reported that the anti-Toxoplasma effect could be confirmed with less expensive metal nanoparticles, specifically iron oxide nanoparticles (goethite and hematite). The manuscript needs to be improved based on my major comments.

[1]   Introduction: The background of the study should be made very clear. Provide more details of the introduction and review of the work.

[2]   Can you explain the mechanism by which iron oxide nanoparticles, specifically goethite and hematite, exhibit anti-Toxoplasma properties?

[3]   How was the surface modification of iron oxide nanoparticles with l-tryptophan confirmed, and what role does this modification play in enhancing their selectivity against T. gondii?

[4]   Could you elaborate on the specific methodology used to assess cytotoxicity and growth inhibition of both naked and l-tryptophan-modified iron oxide nanoparticles?

[5]   What evidence supports the assertion that l-tryptophan-modified nanoparticles exhibit superior anti-Toxoplasma action compared to their naked counterparts?

[6]   Table 1 and 4 provide statistical significance?

[7]   Were any potential off-target effects or unintended consequences observed with the use of l-tryptophan-modified nanoparticles?

[8]   Can you discuss the significance of the finding that the bio-modified nanoparticles did not exhibit detectable host cell toxicity at effective anti-Toxoplasma doses?

[9]   How does the addition of the antioxidant Trolox impact the growth inhibition induced by the bio-modified nanoparticles, and what implications does this have for the role of reactive oxygen species in their anti-Toxoplasma action?

[10]                       Are there any limitations or challenges associated with the use of iron oxide nanoparticles for anti-Toxoplasma therapy that were encountered during the study?

[11]                       What potential advantages or disadvantages do iron oxide nanoparticles possess compared to other metal nanoparticles, such as gold, silver, and platinum, in terms of anti-Toxoplasma efficacy and practicality?

[12]                       How do the findings of this study contribute to the broader field of biomedical research, particularly in the development of novel treatment approaches for parasitic infections like toxoplasmosis?

[13]                       Please speculate on the results. The discussion must improve.

[14]                        In the conclusion section. The authors should add the significance of this research and its potential practical application.

[15]                        The MS English needs to be improved. The article's English must be carefully checked for grammatical errors.

Comments on the Quality of English Language

  The MS English needs to be improved. The article's English must be carefully checked for grammatical errors.

Author Response

(The authors gave the same response as above.)

Round 2

Reviewer 1 Report

Comments and Suggestions for Authors

The authors have correctly addressed my initial concerns; however, the text on lines 205-207 still leads to misunderstandings. The authors may consider changing it to: Strong toxicity (approximately 90% reduction in cell viability) was observed at a concentration of 50 µg/mL for tryptophan-coated goethite, whereas concentrations up to 2.0 µg/mL of tryptophan-coated hematite resulted in a 20% reduction in cell viability.

Reviewer 3 Report

Comments and Suggestions for Authors

Requested corrections were completed.